# Comparative Analysis of *In-House* RT-qPCR Detection of SARS-CoV-2 for Resource-Constrained Settings

**DOI:** 10.3390/diagnostics12112883

**Published:** 2022-11-21

**Authors:** Yesit Bello-Lemus, Marco Anaya-Romero, Janni Gómez-Montoya, Moisés Árquez, Henry J. González-Torres, Elkin Navarro-Quiroz, Leonardo Pacheco-Londoño, Lisandro Pacheco-Lugo, Antonio J. Acosta-Hoyos

**Affiliations:** Centro de Investigaciones en Ciencias de la Vida, Universidad Simón Bolívar, Barranquilla 080002, Colombia

**Keywords:** SARS-CoV-2 diagnostics, RT-PCR, *in-house* PCR

## Abstract

We developed and standardized an efficient and cost-effective *in-house* RT-PCR method to detect severe acute respiratory syndrome coronavirus 2 (SARS-CoV-2). We evaluated sensitivity, specificity, and other statistical parameters by different RT-qPCR methods including triplex, duplex, and simplex assays adapted from the initial World Health Organization- (WHO) recommended protocol. This protocol included the identification of the E envelope gene (E gene; specific to the *Sarvecovirus* genus), RdRp gene of the RNA-dependent RNA polymerase (specific for SARS-CoV-2), and RNase P gene as endogenous control. The detection limit of the E and the RdRp genes were 3.8 copies and 33.8 copies per 1 µL of RNA, respectively, in both triplex and duplex reactions. The sensitivity for the RdRp gene in the triplex and duplex RT-qPCR tests were 98.3% and 83.1%, respectively. We showed a decrease in sensitivity for the RdRp gene by 60% when the E gene acquired Ct values > 31 in the diagnostic tests. This is associated with the specific detection limit of each gene and possible interferences in the protocol. Hence, developing efficient and cost-effective methodologies that can be adapted to various health emergency scenarios is important, especially in developing countries or settings where resources are limited.

## 1. Introduction

Coronavirus disease 2019 (COVID-19), associated with a severe acute respiratory syndrome, is caused by the novel Severe Acute Respiratory Syndrome Coronavirus 2 (SARS-CoV-2) responsible for the current pandemic, first identified in Wuhan, China at the end of 2019 [1,2,3,4]. The infection caused by this new virus triggers a critical respiratory disease [5,6] and has resulted in more than 625 million cases and more than 6.56 million deaths worldwide, according to the data obtained from the World Health Organization as of 25 October 2022 [7,8].

The timely identification of patients who have been infected with SARS-CoV-2 is critical for the relevant public health interventions, epidemiological control, and their articulation for managing the policies that help control the pandemic [9,10]. Following infection, SARS-CoV-2 replicates its genetic material inside the host cell; hence, the gold-standard diagnostic test for detecting infection is through the detection of the viral genetic material using the reverse transcription-coupled real-time polymerase chain reaction technique (RT-PCR) [11,12,13]. More than 350 protocols and commercial kits have been developed for detecting viral genetic material by selecting different regions of the viral genome as targets for amplification and detection [14]. However, the envelope gene (E gene; specific to the *Sarvecovirus* genus) and the RNA-dependent RNA polymerase (RdRp gene specific for SARS-CoV-2) have been widely used as gene-amplification targets for the detection of this virus since they are regions of low variability and therefore present a high probability of detection (>95%) in the diagnostic assay for SARS-CoV-2, resulting in the detection of low viral concentration in a sample, as low as 5.2 and 3.8 copies per reaction, respectively [15,16,17], Pan American Health Organization (PAHO) suggest that only the E gene could be used in places where circulation of SARS-CoV-2 has been established or in countries where no other *Sarbecovirus* are present, such as in South America [18,19,20]. However, the detection sensitivity of these genes decreases with the decrease in the patients’ viral load over time and depends on the quality of the biological sample, thus generating false negatives [21]. Hence, RT-qPCR protocols for the diagnosis of SARS-CoV-2 should be validated and reassessed to ensure their accuracy, considering varying characteristics under the conditions of each laboratory. In this article, we aimed to evaluate the detection limits, specificity, sensitivity, and accuracy of diagnosing SARS-CoV-2 through fast and cost-effective *in-house* RT-qPCR protocols by implementing multiple genes in a single reaction to detect the *Sarvecovirus* genus and confirm SARS-CoV-2 by evaluating both the E and RdRp genes.

## 2. Materials and Methods

### 2.1. Detection Limit Assay

A total of 200 μL from nasopharyngeal swab samples received during the epidemiological surveillance at the Genetics and Molecular Biology Laboratory of Universidad Simón Bolívar that had previously been detected as positive for SARS-CoV-2 using the GeneFinder™ COVID-19 Plus RealAmp Kit (OSANG Healthcare Co., Ltd., Suwon, Korea), which detects E, RdRp, and N viral genes, were selected for the study. Viral RNA was extracted with magnetic beads using the MGIEasy Nucleic Acid Extraction Kit (MGI Tech Co., Shenzhen, China Korea), following the supplier’s protocol. RNA was eluted in 50 μL of ultrapure water. Subsequently, eight serial dilutions of the total extracted RNA were prepared, and the viral load for the study sample was calculated based on a standard curve and by relating the Ct value with the number of viral copies (Appendix A). The values for the standard curve were obtained by the serial dilution of a 3180 pb pUC series plasmid at a concentration of 10 ng/µL, containing specific sequences for the viral E and RdRp genes equivalent to 2.86 × 10^9^ viral copies per µL of the sample (the plasmid was donated by Jaime Castellanos, Virology Laboratory, Universidad del Bosque). The initial viral copies were calculated as a function of the initial plasmid concentration, bp length, and %GC (50%), according to the following equation: Copy number = (ng × number/mol)/(bp × ng/g × g/bp mol). The number of viral copies for the calibration curve (x value) was determined using the straight-line equation (y = mx + b) with R^2^ = 0.993 (Appendix A). The calibration curve was verified using Genefinder^TM^ kit as a commercial kit to compare with our duplex reaction (Appendix A).

From each of the RNA dilutions, the triplicate detection of SARS-CoV-2 was conducted using RT-qPCR with a CFX96™ PCR real-time detection system (BioRad) using the SuperScript™ Platinum™ One Step SuperScript III Platinum RT-qPCR Kit (Invitrogen); TaqMan primers and probes for the E genes (specific to the *Sarvecovirus* genus), RNA-dependent RNA polymerase RdRp genes (specific for SARS-CoV-2), and RNase P genes as an endogenous control were synthesized and provided by BIOSEARCH^TM^ Technologies [15]. Three parallel RT-qPCR reactions were performed: triplex^E:RdRp:RNaseP^ (E+RdRp+RNase P), duplex^E:RNaseP^ (E+RNase P), and duplex^RdRp:RNaseP^ (RdRp+RNase P), for measuring the detection limits and performance characteristics. For the triplex reaction, a mixture of 20 µL was prepared (5 µL sample, 10 µL 2 × reaction buffer from the SuperScript™ Platinum™ One Step SuperScript III Platinum RT-qPCR Kit, 0.5 µL reverse transcriptase/Taq mixture, and 4.5 µL primers and probes); for the duplex reaction, a 16-µL mixture was prepared (5 µL sample, 8 µL SuperScript™ 2 × reaction buffer, 0.3 µL reverse transcriptase/Taq mixture, and 2.7 µL primers and probes). The sequences and concentrations of the primers and probes are described in Table 1. The duplex RT-qPCR protocol parameters were adjusted in four steps: 50 °C for 15 min followed by 95 °C for 3 min and ending with 45 cycles of 95 °C for 15 s and 58 °C for 30 s. The real-time reaction protocol was adjusted for triplex RT-qPCR in four steps: 50 °C for 20 min followed by 95 °C for 3 min and ending with 45 cycles of 95 °C for 15 s and 58 °C for 60 s. The cutoff point for the Ct value was 40 (Table 2). We defined the Ct value where the relative fluorescence exceeded a threshold, set according to the thermal cycler-assigned signal-to-noise ratio (S/N) [22].

### 2.2. Sensitivity, Specificity, and Accuracy Assay

This evaluation included a primary study of diagnostic tests where the operating characteristics were estimated in terms of sensitivity, specificity, negative predictive value (NPV), positive predictive value (PPV), and accuracy of RT-qPCR triplex^E:RdRp:RNaseP^ and duplex^RdRp:RNaseP^ versus RT-qPCR duplex^E:RNaseP^ tests as standard diagnostic tests for detecting SARS-CoV-2 in nasopharyngeal swab samples that were obtained from 132 Colombian patients, of whom 59 were positive for SARS-CoV-2 and 73 were negative, detected using the RT-qPCR protocol based on Corman et al. [15]; samples were received and kept at −80 °C for less than 24 h, and viral RNA was extracted with magnetic beads using the MGIEasy Nucleic Acid Extraction Kit (MGI Tech Co., Shenzhen, China) following the supplier’s protocol. RNA was eluted in 50 μL of ultrapure water, aliquoted in 10 µL, and kept at −80 °C.

### 2.3. Statistical Analysis

The Ct values and viral load of the serial dilutions for SARS-CoV-2 in the triplex and duplex RT-qPCR amplifications were compared through paired and independent *t*-tests, with *p* < 0.05 and 95% confidence using STATGRAPHICS Centurion XVI Versión 16.2.04 software. The sensitivity, specificity, accuracy, and negative and positive predictive values were obtained using Excel software and employing the protocol provided by Sebastián Bravo-Grau et al. [23].

## 3. Results

We evaluated the limit of detection of SARS-CoV-2-positive samples with a Ct of 15 in duplex and triplex reactions. Our results show that the detection limit of the E gene was up to a dilution of 1 × 10^−7^ of the RNA sample, equivalent to 3.3 viral copies per µL of RNA in both triplex and duplex RT-qPCR assays (Figure 1A,D). For the detection of the RdRp gene, the limit of detection was reduced up to a dilution of 1 × 10^−6^, equivalent to 33.8 viral copies per µL of RNA, in both triplex and duplex assays (Figure 1B,C), suggesting that the reduction in detection of the RdRp gene is not due to interference by the E gene, but a decrease in specificity by the RdRp primers. Our limit of detection assays showed a 10× increase in sensitivity for the E gene over the RdRp gene, reflected in a difference in the Cts of about four cycles lower for every dilution for the E gene (Figure 2).

The Ct values from the RT-qPCR triplex test detecting the E and RdRp genes together had significant differences within the assay(*p* = 0.030), as did those from the independent detection of the E and RdRp genes by the duplex RT-qPCR (*p* = 0.009, 95% confidence; Figure 3). However, the Ct values of the detection of the RdRp gene by triplex and duplex RT-qPCR independently did not show significant differences (*p* = 0.174, 95% confidence; Figure 3). Likewise, no significant difference in the analysis of the paired samples with the E genes, duplex vs. triplex, was observed (*p* = 0.410, 95% confidence; Figure 3).

The sensitivity of a diagnostic test enables us to relate the accurate proportion of people who have been infected and have been classified as such through the test. For the triplex RT-qPCR test, the sensitivity was calculated to be 98.3%, and for the RT-qPCR duplex^RdRp:RNaseP^ test, the sensitivity was 83.1%. The positive predictive value was 100% for all the cases (triplex and duplex RT-qPCR). The negative predictive value which evaluates the conditional probability that individuals with a negative result do not have the disease was 98.7% for the triplex RT-qPCR test and 87.6% for the RT-qPCR duplex^RdRp:RNaseP^ test. The accuracy or proportion of individuals accurately classified was 99.2% for triplex RT-qPCR and 92.3% for duplex^RdRp:RNaseP^ RT-qPCR tests. All the values for the above parameters were 100% when analyzed between Ct values of 13 and 30 for the E gene (Table 3). In 60% of the positive cases, the RdRp genes were not detected when the E genes acquired Ct values > 30 in the diagnostic tests, whereas for the triplex RT-qPCR, the detection of any of the two genes had a sensitivity of 75% when the Ct was in the 36–38 range (Table 3). We presented a fast and cost-effective protocol giving a detailed comparison of different gene targets and limits of detection showing a higher sensitivity of the E gene over the RdRp gene; however, when we used both genes in the multiplex reaction it resulted in the highest specificity, sensitivity, PPV, and NPV of the assays reported herein.

## 4. Discussion

We standardize a cost-effective and fast RT-qPCR multiplex protocol that can be adapted for different laboratory settings using open diagnostic equipment and platforms, without the need for expensive commercial detection kits. In this study, we reported a false negative rate of 12.4% of the total true positives for the duplex RT-qPCR test detecting the RdRp gene. However, the triplex RT-qPCR test, which detected both E and RdRp genes, decreased the rate of false negatives to 1.3%. These results may be influenced by the quality and integrity of the samples before and after RNA extraction, as they are thermolabile and susceptible to degradation [24], as well as by the loss of specificity or discordance of the oligonucleotides directed to the RdRp genes, as reported in other studies [18]. The sensitivity and negative predictive values for the triplex and duplex tests were consistent with the detection limit of the E and RdRp genes, which have a difference of detection of 10×, thereby differing from the results reported by the Charité-Berlin protocol [15] and being more closely related to those reported by other authors [18,25] and thus requiring the evaluation of novel confirmatory markers with a lower mutation rate or interference between targets. As we managed to reduce the amount of commercial enzyme and master mix up to 50% less than what the manufactured suggested without affecting the performance, we were able to reduce the cost of the assay by half compared to other protocols and with preassembled commercial kits, making our protocol suitable for low-income settings.

The generation of positive samples for the detection of the E gene using RT-qPCR that were negative in the duplex RT-qPCR for the RdRp genes may be associated with a low sensitivity of the RdRp marker, possibly related to a low matching affinity by the reverse primer [18,26]. Likewise, the RT-qPCR reaction created artifacts that generated false negatives or positives, depending on other factors that are independent of the nature of the sample or virus sequence variability [18], such as the thermocycler, open or closed systems [27], diagnostic kits [28], reagent quality [29], and even the consumables, such as the PCR tubes or plates available in the laboratory [22,30]. This is especially relevant when there is a shortage of reagents due to the high demand caused by the health emergency, as observed in the early stages of the current pandemic.

The generation of false negatives in the diagnosis of SARS-CoV-2 reached approximately 67% and 66% before days 4 and 21 following infection, respectively, and the lowest rate of false negatives (20%) was observed on day 8, according to Kucirka et al. [31]. Therefore, to ensure reliable results, RT-qPCR diagnostic tests for SARS-CoV-2 should be validated in terms of reproducibility of the results with standard tests, which display the accuracy of such tests that need to be adopted in the laboratory, as well as careful interpretation of the results at the initial stages of the viral infection.

In conclusion, this study presents detailed comparison of RT-PCR methods that can be cost-effective for the detection of SARS-CoV-2. We showed the importance of establishing fast and economical multiplexed assays that can be applied under different lab conditions depending on the availability of reagents. Finally, accurate and prompt detection of SARS-CoV-2 or other relevant public health pathogens under different laboratory settings is crucial for mitigating current or new epidemics.

## Figures and Tables

**Figure 1 diagnostics-12-02883-f001:**
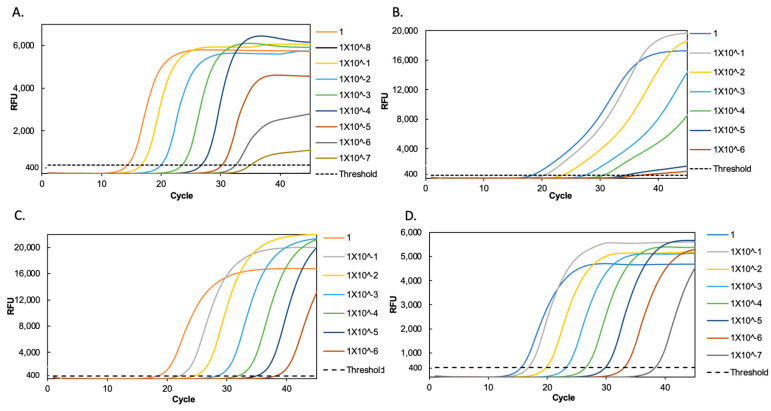
Amplification curves of RT-qPCR for the different assays. (**A**) E marker in triplex RT-qPCR, (**B**) RdRp marker in triplex RT-qPCR, (**C**) RdRp marker in duplex RT-qPCR, and (**D**) E marker in duplex RT-qPCR, all for different dilutions of viral RNA from a SARS-CoV-2 positive sample.

**Figure 2 diagnostics-12-02883-f002:**
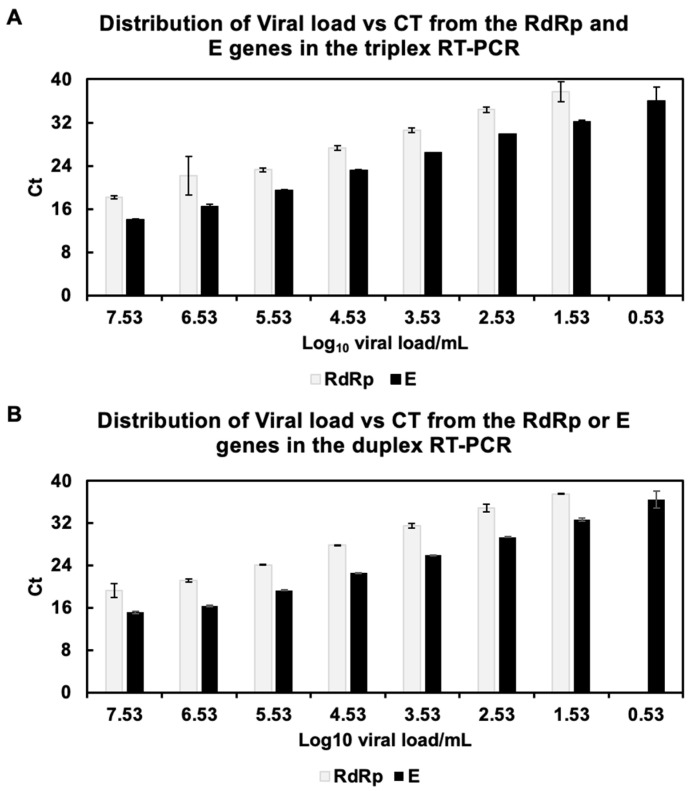
Limit of detection of RdRp and E genes in (**A**) duplex RT-PCR or (**B**) triplex RT-PCR. The E gene shows more sensitivity than the RdRp gene, both in the duplex and triplex assays. Average Ct values and SD from triplicate reactions are shown.

**Figure 3 diagnostics-12-02883-f003:**
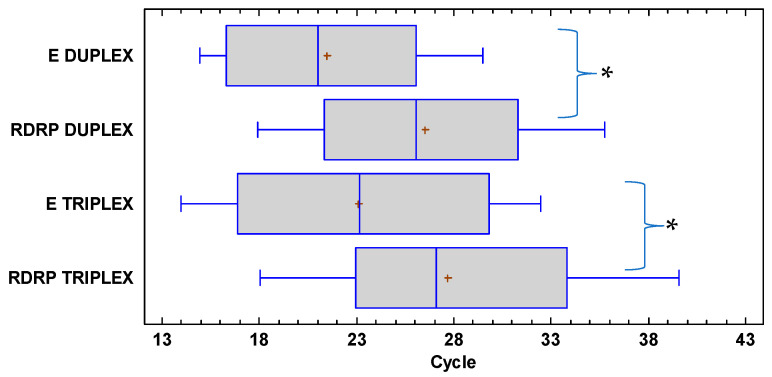
Whisker plot showing the distribution of the median and quartile values distributed between the triplex and duplex RT-qPCR treatments in the detection of the E and RdRp genes. * *p* < 0.05 between the Ct of both genes.

**Table 1 diagnostics-12-02883-t001:** List of primers and probes for E, RdRp, and RNase P genes.

Gene Name	Oligonucleotide ID	Sequence (5′–3′) [15]	TRIPLEX * ^E:RdRp:RNase P^	DUPLEX * ^E:RNase P^	DUPLEX * ^RdRp:RNase P^
RdRp	Forward	GTGARATGGTCATGTGTGGCGG	600 nM	-	600 nM
	Reverse	CARATGTTAAASACACTATTAGCATA	800 nM	-	800 nM
	Probe_P2	FAM-CAGGTGGAACCTCATCAGGAGATGC-BBQ-1	200 nM	-	200 nM
E	Forward	ACAGGTACGTTAATAGTTAATAGCGT	400 nM	200 nM	-
	Reverse	ATATTGCAGCAGTACGCACACA	400 nM	200 nM	-
	Probe_P1	CAL FLUOR RED 610-ACACTAGCCATCCTTACTGCGCTTCG-BBQ-2	100 nM	200 nM	-
RNase P	Forward	AGATTTGGACCTGCGAGCG	200 nM	100 nM	100 nM
	Reverse	GAGCGGCTGTCTCCACAAGT	200 nM	100 nM	100 nM
	Probe_Pro1	HEX-TTCTGACCT-Nova-GAAGGCTCTGCGCG- BHQ-1	100 nM	100 nM	100 nM

* Values correspond to Final Concentrations.

**Table 2 diagnostics-12-02883-t002:** Components for the preparation of the triplex and duplex RT-qPCR master solution for detecting viral E, RdRp, and RNase P genes as internal control for the diagnosis of SARS-CoV-2.

		RT-qPCR TRIPLEX	RT-qPCR E DUPLEX	RT-qPCR RdRp DUPLEX	
Component	Stock Concentration	Volume per 20 μL Reaction	Volume per 16 μL Reaction	Volume per 16 μL Reaction	Final Concentration
2 × SuperScript™ RB	-	10	8	8	-
RT/Taq	-	0.5	0.3	0.3	-
Forward primer RdRp	48 µM	0.25	-	0.2	0.6 µM
Reverse primer RdRp	64 µM	0.25	-	0.2	0.8 µM
Forward primer E gene	32 µM	0.25	0.2	-	0.4 µM
Reverse primer E gene	32 µM	0.25	0.2	-	0.4 µM
Forward primer RNase P	16 µM	0.25	0.2	0.2	0.2 µM
Reverse primer RNase P	16 µM	0.25	0.2	0.2	0.2 µM
Probe_P2 RdRp	4 µM	1	-	0.8	0.2 µM
Probe_P1 E	2 µM	1	0.8	-	0.1 µM
Probe_Pro1 RNase P	2 µM	1	0.8	0.8	0.1 µM
RNA (add at step 4)	-	5	5	5	-
H_2_0 nuclease free	-	-	0.3	0.3	-

**Table 3 diagnostics-12-02883-t003:** Relationship of the sensitivity, specificity, accuracy, and positive and negative predictive values between the Ct heat ranges of the standard test for the RdRp duplex RT-qPCR and RdRp Triplex E RT-qPCR diagnostic assays, including 59 positive samples and 73 negative samples.

Duplex^RdRp:RNaseP^							
	Total	Ct 13–15	Ct 16–20	Ct 21–25	Ct 26–30	Ct 31–35	Ct 36–38
Sensitivity (%)	83.1	100	100	100	100	68.2	54.5
Specificity (%)	100	100	100	100	100	100	100
PPV (%)	100	100	100	100	100	100	100
NPV (%)	87.6	100	100	100	100	91.3	93.6
Accuracy (%)	92.3	100	100	100	100	92.6	94.0
**Triplex^E:RdRp:RNaseP^**							
Sensitivity (%)	98.3	100	100	100	100	100	75.0
Specificity (%)	100	100	100	100	100	100	100
PPV (%)	100	100	100	100	100	100	100
NPV (%)	98.7	100	100	100	100	100	97.3
Accuracy (%)	99.2	100	100	100	100	100	97.4

## Data Availability

All data presented here are available upon request to authors.

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
