# Peer review of "Comparative Analysis of In-House RT-qPCR Detection of SARS-CoV-2 for Resource-Constrained Settings"

_diagnostics, 2022, doi:10.3390/diagnostics12112883_

Round 1
Reviewer 1 Report
The study describes the comparison of real-time PCR methods for the detection of SARS-CoV-2.
The study conducted by the authors are very important and maybe useful for rapid detection of the virus. However, I have few issues for the manusript, which should be considered by the Bello-Lemus et al.:
1. Line 56-71- including the references is needed here
2. The further information of the STATGRAPHICS Centurion XVIII software should be included in this part
3. The authors used few time in manucript expresion „economical”. Could you explain it regarding to in house RT-PCR method in disccussion?
Author Response
|
REVIEWER 1 |
ACTION |
|
The study conducted by the authors are very important and maybe useful for rapid detection of the virus. However, I have few issues for the manusript, which should be considered by the Bello-Lemus et al. |
|
|
1 Line 56-71- including the references is needed here |
We included references. |
|
2 The further information of the STATGRAPHICS Centurion XVIII software should be included in this part |
We accepted the comment, and we included a more detailed description. |
| 3 The authors used few time in manucript expression “economical”. Could you explain it regarding to in house RT-PCR method in disccussion? | We added a paragraph in discussion where we comment about the cost of the assay. We changed “economical” for “cost-effective” in several lines throughout the manuscript. |
Reviewer 2 Report
Comments to the Author
In the current study, the authors reported the standardization of duplex and triplex real-time PCR for SARS-CoV-2. The first objective of this work is to optimize and permit assessment of the in-house methods for SARS-CoV-2 diagnosis. The primers and probes used by real-time reactions were described before by Corman et al., 2020. It is interesting to encourage different research groups to publish their experiences and difficulties encountered with the SARS-CoV-2 diagnostic. The importance of this work is related to the process of validation of in-house SARS-CoV-2 triplex real-time reaction for future application in the diagnosis of infected individuals and for being considered as a solution instead of commercial molecular assays. However, this study doesn't add much contribution to the literature because the authors use a protocol already described before, the number of samples evaluated is very small and important restructuration in the article is necessary. In this way, a more compact account, such as a letter, would be more appropriate. In addition, the authors could further explore the results, comparing thoroughly the in-house with gold standard commercial assay describing the differences in the PCR efficiency, Ct value and all the parameters necessary for valuation according to regulatory policies. To report the difficulties encountered and compare with the proposed methodology, highlighting the costs and throughput capacity of each methodology.
Major comments
1) It’s not clear the advantage to use duplex reaction E gene + RNase P once this gene is not specific for SARS-CoV-2 diagnosis. According to CDC for SARS-CoV-2 molecular diagnosis is recommended use of different viral genes and plus an internal control.
2) The article necessities of an important restructuration of the paragraphs and sentences. There is very long sentences and paragraph completely out of context, for instance page 3 lines 97 -105. A language revision also is important.
3) In Material and Methods, it is interesting to describe how the number of samples used for validation of the molecular assay were defined. According to the work of Smith and col. For validation of real-time PCR are necessary complex stages, including the realization of different experiments.
4) It is not clear if the samples were freshed or have been stored before the extraction? After the extraction, were the samples evaluated for Gene Finder and in-house reaction concurrently?
5) In all cases it is important to compare with the gold standard PCR, including the value of viral load and limited detection.
6) The authors could explain better how the viral load was calculated. Are the results based on µl of RNA? Or based on a sample extracted from 200 µl of nasopharyngeal swab? How exactly was calculated the viral load? How was the Ct value for the analysis defined?
7) In the results the sentence “the limit of detection of a SARS-CoV-2-positive samples with a Ct of 15” could be better explained. The authors selected positive samples previously evaluated by GeneFinder that presented Ct value of 15?
8) The table 3 showed part of the validation process using duplex and triplex reactions. How was defined the Ct value? What the number of samples including into the groups of the different Cts? It is surpreendente that the authors found better results for triplex reaction compared to duplex reaction. False negative rate of 12,4% is considered very high.
Minor comments
Include the fabricant of GeneFinder
Table 1. Reference Corman 2020
Include the number of protocol code in the Institutional review Board Statement
Author Response
Please check it in the attachment.
